# Electrostatic Charge Retention in PVDF Nanofiber-Nylon Mesh Multilayer Structure for Effective Fine Particulate Matter Filtration for Face Masks

**DOI:** 10.3390/polym13193235

**Published:** 2021-09-24

**Authors:** Dong Hee Kang, Na Kyong Kim, Hyun Wook Kang

**Affiliations:** Department of Mechanical Engineering, Chonnam National University, 77 Yongbong-ro, Buk-gu, Gwangju 61186, Korea; kdh05010@gmail.com (D.H.K.); naky0607@gmail.com (N.K.K.)

**Keywords:** multilayer structure, PVDF, nanofiber, nylon, electrostatic interaction, triboelectricity, piezoelectricity, face mask

## Abstract

Currently, almost 70% of the world’s population occupies urban areas. Owing to the high population density in these regions, they are exposed to various types of air pollutants. Fine particle air pollutants (<2.5 μm) can easily invade the human respiratory system, causing health issues. For fine particulate matter filtration, the use of a face mask filter is efficient; however, its use is accompanied by a high-pressure drop, making breathing difficult. Electrostatic interactions in the filter of the face mask constitute the dominant filtration mechanism for capturing fine particulate matter; these masks are, however, significantly weakened by the high humidity in exhaled breath. In this study, we demonstrate that a filter with an electrostatically rechargeable structure operates with normal breathing air power. In our novel face mask, a filter membrane is assembled by layer-by-layer stacking of the electrospun PVDF nanofiber mat formed on a nylon mesh. Tribo/piezoelectric characteristics via multilayer structure enhance filtration performance, even under air-powered filter bending taken as a normal breathing condition. The air gap between nanofiber and mesh layers increases air diffusion time and preserves the electrostatic charges within the multi-layered nanofiber filter membrane under humid air penetration, which is advantageous for face mask applications.

## 1. Introduction

Air filtering using nanofiber (NF) membranes is widely applied to industries for air-quality management such as medical supplies, chemical products, and semiconductor manufacturing [1,2]. Comparing with a microfiber membrane, the nanofiber membrane promises an enhanced filtration performance for sub-micron contaminants in the identical pressure drop [3]. In particulate matter (PM) filtration, the NF membrane is used for a high-efficiency particulate arresting grade filter (filtration efficiency (*η*) > 99.97% @ PM 0.3) for air purifiers and an N95-rated respirator (*η* > 95.0% @ PM 0.1–0.3) for face masks [4,5,6,7,8]. However, there is a trade-off between the filtration efficiency and pressure drop. In a filter with a high-pressure drop, high energy consumption requires the supply of highly purified air [9,10].

In PM filtration via a fibrous filter, the 0.3 μm size of PM has most penetrating characteristics, depending on the particle capture mechanism [11,12]. PM with size at 0.3 μm is mainly captured on the NF surfaces via electrostatic interactions during Brownian motion along the flow streamlines inside the membranes. Electrostatic force on the filter surface is vulnerable to exposure of polar gas molecules, moistures and charged particles in the air [13]. To increase the filtration efficiency without geometrical variation of the filter, electrical energy is supplied to the NF surfaces for capturing the fine PM. In previous studies, for the generation of electrostatic interactions on the filter, a metal layer-coated microfiber structure was used to apply an electrical charge from an external source. One research group suggested an aluminum coated polyester microfiber fabrication method for a high-performance electrostatic filtration system [14]. Another research group presented a copper layer-coated polyacrylonitrile microfiber for PM filtration with a decreasing carbon monoxide concentration [15]. The metal layer-coated microfiber membrane demonstrated a pressure drop of less than 10 Pa, which had a direct effect on the improvement of the filter quality factor (QF). In order to maintain high PM filtration efficiency, a voltage of several kilovolts should be continuously supplied to the filter. In addition, the ozone generation rate increases because it reacts with oxygen molecules passing through the filter in proportion to the intensity of the electric field [16]. Another method to apply electrostatic force on the filter surface included embedding of functional group-attached nanoparticles or nanoplates on the NF surfaces [17,18]. The other method, surface modification by plasma treatment, was performed for post-treatment of NF membranes to generate electrostatic charges [19]. The generation of electrostatic forces on the NF surface can increase filtration performance in the membrane fabrication process or post-process. As mentioned for the above methods, the electrostatic force via the plasma treatment or the embedding functional group on the NF membrane surface are only temporarily effective during the first use of a filter, and then the surface potential decreases exponentially on the filter surface exposed to air.

Recently, piezoelectric materials or triboelectric effects have been applied to NF filters to enhance filter performance without external power, via quasi-permanent electric charge or polarization on NFs. The NF membrane filter based on a piezoelectric polyvinylidene difluoride (PVDF) material has self-powered characteristics with the advantage of electric charges on the NF surface. Even when the filter is in use, the electrostatic charge density is improved by the vibration and bending of the nano-net structure on the NFs [20,21]. An NF filter using triboelectric charges was applied to fine PM filtration via mechanical friction and deformation between the polytetrafluoroethylene and nylon materials [22,23]. In this study, an electrostatic air filter for face masks shows self-electrostatic charge generation and retention characteristics in the cyclic air blowing and humid air penetration conditions. A filter membrane is fabricated by electrospinning of the PVDF on a nylon mesh layer, used as a NF support structure. The fine PM filtration efficiency of the filter is compared depending on the electrostatic charge generation and retention characteristics via tribo/piezoelectricity. The filter membrane stacks are composed of nylon mesh layers with the one to two, three, and five layers of electrospun PVDF NF, respectively. It has a structure in which the membrane composed of the PVDF NF-nylon mesh layer is stacked in order. The number of nylon mesh layers of the filter membrane is adjusted in order to enhance the PM filtration rate and diminish the low pressure drop. Increasing the number of the mesh layers in the filter membrane is effective for filtration performance as the PM size decreases. A triple-layer NF filter structure has enhanced electrostatic charge generation on the filter in a respiration mimicking environment, which has advantages for a face mask application. A new type of face mask is possible to generate electrostatic charges in the breathing process after wearing a face mask via the unique multilayer structure of the filter, which overcame the limitation to fine PM filtration performance on the face mask.

## 2. Materials and Methods

For the membrane filter fabrication, the electrospinning method is performed using PVDF solution on the nylon mesh substrate. The multilayer structure of the face mask filter is composed of the stacked electrospun PVDF NF-nylon mesh substrates. The nylon mesh substrate (woven Nylon 66A screen, Flon ind.) has a 161 μm pore size, 40 g m^−2^ area density and 433 cm^−1^ mesh threads. The nylon mesh supporting layer has almost no effect to increase the airflow resistance (~0.5 Pa per nylon mesh layer at 18.64 cm s^−1^ air velocity in the Appendix A). The nylon mesh layer was cut into a 100 × 100 mm^2^ areas and washed sequentially for 5 min each using acetone, ethanol, and deionized (DI) water in an ultrasonic cleaner. The washed mesh was dried in a convection oven (OF-12PW, Jeio tech) for 12 h at 60 °C. The PVDF NF membrane was fabricated by electrospinning using 13 wt% PVDF (M_w_~534,000), which was dissolved in acetone and n-dimethylformamide (DMF) mixed solvents at a 7:3 volume ratio. All chemicals in the PVDF solution were purchased from Sigma-Aldrich (analytical grade) and used without further purification. The PVDF solution was homogeneously mixed using an ultrasonicator for 72 h at 50 °C and stirred using a hotplate stirrer at 40 °C and 400 rpm for a week. The PVDF solution was left for an hour to cool to room temperature before electrospinning.

Geometric analysis of the bulk film and NF membrane was performed by scanned electron microscopy (SEM) (JSM-7500F, JEOL, Tokyo, Japan) and surface profiling (NV-E1000, Nanosystem, Ascoli Piceno, Italy). Spectroscopic analysis was performed by X-ray diffraction (XRD) (X’Pert PRO, PANalytical, Malvern, UK) and by Fourier-transform infrared (FT-IR) (Spectrum 400, PerkinElmer, Melville, NY, USA) patterns of the PVDF with NF membrane, bulk and powder. Filtration performance measurement was performed in a chamber (indicated by the pneumatic symbol in Appendix A). Incense was used for the PM source in the first chamber. The generated PM was diluted in the second chamber for decreased concentration and homogeneous PM distribution by mixing of pumped air. In the upstream and downstream of the filter holder, PM counters (DT-9881m, CEM Inst Co., Shenzhen, China) and pressure sensors (testo 510, Testo SE & Co., Titisee-Neustadt, Germany) were located to measure the filtration efficiency and the pressure drop of the filter. The quality factor was calculated by using results of the filtration efficiency and pressure drop.

The experimental apparatus was prepared for the electrical discharging of the filter membrane via humid air penetration (Appendix A). The moisture concentration in the supplied air was calculated by the average absolute humidity per minute as a level of typical breathing for an adult (Appendix A). In the mask filtration test, the filter membrane is electrically charged by cyclic air blowing (Appendix A). The mask is composed of the polypropylene (PP) layer-covered filter membrane, which was deformed by cyclic air blowing at 15 cycles min^−1^. The power of the airflow was calculated based on the maximum flow rate and pressure difference of typical breathing for a healthy adult (Appendix A).

## 3. Results and Discussion

### 3.1. Materials for Electrostatic Filter

The electrostatic force on the filter surface is more dominant when the PM size is smaller. However, it exponentially disappears on the NF surfaces due to the electrostatic interaction in the air with charged particles and humid molecules. Electrostatic charges on the NF surface should be generated as much as disappeared to preserve the filtration performance. In the fibrous membrane filter for face mask application, tribo- or piezo-electric interactions in the filter are suitable for self-powered electrostatic charge generation, even while in use. A mask filter undergoes to the continuous mechanical deformation by the inhalation-exhalation cycles during the PM filtration. From this point of view, we propose to fabricate a filter membrane with a structure designed as a triboelectric generator involving piezoelectric material; then the filter membrane is composed with PVDF NF structure and nylon mesh supporting layer.

Mesh layer flattening was performed on the electrically grounded aluminum foil collector to maintain a uniform electric field for NF membrane fabrication as shown in Figure 1a. A high voltage source (B.150, KSC Korea switching, Korea) was used to produce the electrical potential gradient (15 kV) on the aluminum foil collector leading from the metal needle tip (needle tip-to-collector distance, 150 mm). The Figure 1b shows that the positively charged PVDF solution (flow rate, 30 μL min^−1^) stretches to the NF geometry and randomly piled up on the mesh. During the electrospinning process, the acrylic cover layer works as an insulation layer in the electric fields. The electrospun nanofiber was formed in a circular shape on the mesh substrate, which was not covered by the acrylic layer. Depending on the fabrication conditions, each membrane for the experiment was prepared with 10 samples. The PVDF NF layer was attached to the mesh layer through dry adhesion by van der Waals forces due to the momentarily induced dipole moment of the polymer chain during the electrospinning process [24]. The mesh layer was used as the substrate to accumulate the PVDF NF layer in the electrospinning and to support the NF layer from physical damage in the filter membrane. After PVDF NF layer fabrication on the mesh layer as shown in Figure 1c, it was dried in the oven for 24 h at 40 °C to remove remaining solvent on the NF layer. A filter membrane with a single-layer PVDF NF was completed by covering another nylon mesh layer on the PVDF NF-nylon mesh layers. For the fabrication of the filter membrane with a multilayer of PVDF NFs, the mesh layer divided a PVDF NF layer into two to three, and five layers by the number of mesh layers, which were stacked for construction of a multilayer structure as shown in Figure 1d. The total electrospinning time to form PVDF NF was maintained to 900 s in each multilayer filter membrane. Electrostatic charges were generated on the nylon mesh and NF membrane surfaces by physical friction and mechanical deformation via bending, fluttering, and vibrating of the filter. (Figure 1e) The two materials are far apart in the triboelectric series to easily generate electrical charges on their surfaces by physical friction [25]. The PVDF has piezoelectric characteristics which generate electric charge in mechanical deformation.

### 3.2. Geometric and Spectroscopic Analysis

The electrostatic characteristics of the PVDF film are analyzed based on the geometry of the bulk film and NF membrane. The geometric characteristics of the PVDF film are analyzed based on the SEM images in Figure 2a,b. As shown in Figure 2(a-1,a-2), the bulk film has a uniform surface with 45.11 ± 1.97 μm thickness. The weight measurements of the bulk film per unit area are compared by multiplying the PVDF density and the average thickness value from the surface profiler (Appendix A); a difference of less than 1.8% is observed. The NF membrane thickness is adjusted to match the thickness of the bulk film by controlling the electrospinning time. In Figure 2(b-1,b-2) the membrane layer shows entangled NFs of various diameters (260.50 ± 148.80 nm, Appendix A) forming a porous structure with 44.48 ± 1.02 μm thickness.

In the electrospinning process for PVDF NF membrane fabrication, the polymer chain of the α-phase is changed to the β- or γ-phases in the PVDF crystalline structure. The α and β phases are crystalline domains, which are formed by a mixture of TGTG’ (Trans-Gauche-Trans-Gauche conformations) chains and all-TTTT trans planar zigzag chains, respectively [26]. The induced dipole moment is aligned by electrical poling in a strong electric field strength (0.1 kV mm^−1^). Mechanical stretching also affects the arrangement of dipoles by elongation of the PVDF solution to the NF [27,28]. These results are reflected in the XRD and FT-IR patterns of the NF membrane, as shown in Figure 2c,d. In the FT-IR spectra, the relative absorbance intensity of the β-phase in the PVDF NF membrane is notable at 1262, 1072, and 839 cm^−1^ compared to that of the PVDF bulk film or the PVDF powder [29]. The XRD pattern shows the intensity ratio of each PVDF film and powder with 2θ ranging from 5 to 50°. The PVDF powder shows a typical diffraction peak of the α-phase crystalline structure at 2θ = 18.37°(020), 19.93°(110), and 26.58°(021), corresponding to each diffraction plane. The bulk PVDF film presents a developed γ-phase at 2θ = 20.32°(110, 101). The PVDF NF membrane shows relatively strong peaks due to the β-phase crystalline structure at 2θ = 20.68°(110, 200) [30]. The improved β-phase ratio in the PVDF crystalline structure enhances the electrostatic charges on NF surfaces via piezoelectricity in mechanical NF deformation [31].

### 3.3. Electrostatic Characteristics of the PVDF NF Layer

The electrostatic properties of PVDF films, depending on their geometric shape, were analyzed as normalized surface potentials (*V*/*V*_0_) on the PVDF films through discharge on the surface. The electrical potential over time (*V*) on the film surface is divided by the initial electrical potential (*V*_0_). A friction cycle with contact-separation motion (100 times for 120 s, 0.83 Hz) between the PVDF film and nylon mesh causes the formation of electrostatic charges on the NF membrane and bulk film surfaces. The normalized surface potential in Figure 3a shows an exponential decay, which is expressed by Equation (1):(1)$$\frac{V}{V_{0}}=e^{-\frac{t}{\tau }}$$
where *t* is the time, and *τ* is the time constant of the exponential decay of the surface voltages. The electrical potential on the film surfaces was measured for 1200 s and was fitted to calculate the time constant. The value of 1*τ* for the bulk film and NF membrane is 42.7 s and 6042 s, respectively. Regardless of the PVDF film surface being discharged through direct contact with the ground metal surface, the NF membrane has 140 times higher electrostatic charge retention characteristic until 1*τ* than the bulk film. The PVDF NF membrane has superior electrostatic charge sustainability compared to the bulk film. Voids of the NF membrane trap the electric charges despite the direct contact with the ground state of the metal surface. From Appendix A, the porosity of the NF membrane is calculated to be 92.76% when it has the same volume as the bulk film. In addition to saving raw material for porous NF membrane fabrication, electrostatic charges trapped inside the membrane are retained for a long time owing to the high specific surface area. Based on the superior retention properties of electrostatic charges on the PVDF NF membrane, the PM filtration performance is analyzed depending on the membrane thickness control. The thickness of the PVDF NF membrane is controlled by varying the electrospinning time, and a linear correlation is observed between them (Appendix A). On the single-layer filter, regardless of the NF membrane thickness, the discharged NF membranes show an intrinsic surface potential of PVDF (−13.0 ± 2.95 V), as shown in Figure 3b. For the charged single-layer filter, as the thickness of the NF membrane increases, the magnitude of surface potential increases linearly. The initial surface potential is −85.7 V in the fabricated NF membrane with a thickness of 40 μm.

The pressure drop of the filter layer increased as the NF membrane thickness increased. The PM filtration efficiencies are mainly dependent on the NF membrane thickness. As shown in Figure 3c, *η* at all PM sizes increases with increasing NF membrane thickness. The filtration efficiency (*η*) was calculated based on the number of particles by size (2.5, 1.0, 0.5, and 0.3 μm) using Equation (2):(2)$$\eta =\frac{N_{i}-N_{o}}{N_{o}}$$
where the number of particles *N_i_* and *N_o_* are measured upstream and downstream of the filter, respectively. At PM 0.5, the *η* values of the discharged single-layer filters are 0.30, 0.57, 0.83, 0.85, and 0.92 for NF membrane thicknesses of 7.9, 13.2, 18.5, 26.5, and 39.7 μm, respectively; thus, they increase with increasing thickness of the NF membrane. However, there are physical barriers to improving *η*, which is accompanied by an exponential increase in the pressure drop. Even without NF membrane thickness control, electrostatic charging on the single-layer filter improves the filtration performance of all filters. For the NF membrane with 18.5 μm thickness for a single-layer filter, *η* is dramatically enhanced compared to other single-layer filters thickness conditions; *η* values of 18.79%, 12.76%, 8.78%, and 6.59% are obtained for PM filtering with 0.3, 0.5, 1.0, and 2.5 μm sizes, respectively. In particular, under the PM 0.3 filtration condition, the Coulomb force on the NF surface mainly affects the filter. For the NF membrane with 39.7 μm thickness for a single-layer filter, *η* increases by electrical surface charging to only 6.20%, 6.85%, 3.83%, and 1.62% for the PM filtering with 0.3, 0.5, 1.0, and 2.5 μm sizes, respectively. There is a trade-off relationship between the variables *η* and Δ*P*, which can be expressed by a QF, as shown in Equation (3):(3)$$\mathrm{QF}=\frac{-\ln\left(1-\eta \right)}{\Delta P}$$

Appendix A shows the QFs for various NF membrane thicknesses and charge states for the single-layer filter depending on the PM size. In Figure 3d, the QF shows the effectiveness of single-layer filters with the presence of electrostatic charges on the filter surface for PM 0.3 and 0.5 filtration. The NF membrane with 18.5 μm thickness is most effective in increasing the QF by surface charging. From the analysis of the obtained QFs, the filter layer should have an increasing air penetration time to filter and improve the Coulomb force on the NF surface by the increased filter layer composition. Accordingly, the NF membrane-mesh layers are stacked vertically to fabricate a multilayer structure for effective filtration of fine PM.

### 3.4. Evaluation of Filtration Performance

For the analysis of filtration performance depending on the number of stacked layers, the total electrospinning time is maintained at 900 s to fabricate an NF membrane or membrane in the multilayer filter. As shown in Figure 4a, each multilayer filter consists of one, two, three, and five layers of the NF membrane-mesh, with electrospinning times of 900, 450, 300, and 180 s, respectively. As the number of layers increases from one to two, three, and five, the pressure drop of the multilayer filters increase to 2.88%, 10.1%, and 17.9%, respectively, at 18.65 cm s^−1^ face velocity. The thickness of the quintuple-layer NF membrane multilayer filter is 2.26 times higher than that of the single-layer filter owing to the increased number of mesh layers. Because of the large pore size (161 μm) of the nylon mesh compared to that of the PM, it does not have a significant effect on the variation of pressure drop. The mesh maintains air gaps between the NF membranes and preserves electrostatic charges inside the multilayer filter. As shown in Figure 4b, at the PM 2.5 filtration condition, the *η* values have no variation depending on the multilayer structure. As the PM size decreased, the electrostatic forces on the multilayer filter had significant effects on the fine PM filtration. Despite the same electrospinning time conditions for fabricating NF membranes, *η* increased from 95.7% for a single-layer filter to 99.5% for a quintuple-layer filter at PM 0.3. For the multilayer filter, the increased filter thickness increases the required time for passing particles through the electrostatically charged NF surface, which leads to improved depth filtration characteristic. Compared with the performance of commercial face mask filters (Appendix A), the multilayer filter shows a considerably higher performance in terms of the Δ*P* and *η* characteristics.

Based on the hydrophobic properties of pristine PVDF, the porous NF membrane structure has a superhydrophobic surface, corresponding to a Cassie-Baxter model [32]. An NF surface with low surface energy can be temporarily improved by electrostatic forces, which are induced by mechanical stress on the β-phase crystalline in PVDF (piezoelectricity) and by physical friction between the filter structures of PVDF and nylon (triboelectricity). This makes it possible to easily remove charged impurities adhering to the NF surfaces through intermolecular interactions via the dipole moment of polar liquids such as ethanol or DI water [33]. In Figure 4c, optical microscopy (OM) and SEM images show the NF membrane surface of the multilayer filter (Figure 4(c-1)) before and (Figure 4(c-2)) after PM filtration, and after ethanol dipping (completely immersing for 5 min) and drying (in a convection oven for 12 h at 40 °C) of the filter. In the ethanol dipping process, the electrostatic interaction on a PM-aggregated NF surface rapidly vanished via direct exposure to a polar liquid (Figure 4(c-3)). After drying in a convection oven, the filter layer had no physical deformation and no contaminants remained on the NF surface. Figure 4d shows the average QF for 10 days depending on the multilayer filter structure and PM size. All the multilayer filters are measured for filtration performance after the filter regeneration process every day by ethanol dipping, drying, and surface charging. For the quintuple-layer filter, the QF improved by 25.8%, 28.3%, 22.2%, and 3.91% compared to that of a single-layer filter at PM 0.3, 0.5, 1.0, and 2.5 conditions, respectively. In the long-term filtration test, the QF is not significantly affected by the number of stacking layers for PM 2.5 filtration. In contrast, for *η* below PM 1.0, the QF was improved by over 20% through the filter regeneration steps. As shown in Figure 4e, the filter regeneration step is applied to the quintuple-layer filter for 10 days. The *η* is maintained over 98% at PM 0.3 with a constant pressure drop.

For the face mask application, the feasibility of the electrostatic characteristics in the multilayer filter should be analyzed by estimating *η* and QF in a respiration mimicking environment. The filter of the face mask undergoes periodic deformation by breath and humid air exposure by exhalation. First, the multilayer filter is investigated for the tribo/piezoelectric effect through filter deformation by air power. Airflow occurred periodically over the surface of the multilayer filter, with a frequency and intensity similar to that of human breathing. Second, the *η* of the multilayer filter is determined to examine the performance degradation via exposure to continuous humid or dry air penetration. The multilayer filters are covered with melt-blown polypropylene (PP) membranes, which are generally used in the outer and inner layers of commercial disposable face masks.

The schematic in Figure 5a shows that the PP layer-covered multilayer NF filter membrane of the mask is deformed by cyclic air blowing at 15 cycles min^−1^. The QF of the mask is determined based on the number of PVDF NF layers and the charging state. On the left side of Figure 5b, there are no differences in QF regardless of the number of NF layers on the discharged state. After mask bending with cyclic air blowing for 10 min, the QF increased for all filter membranes. The function of the filter membrane with double-layer NF remarkably improves for fine PM filtration. The QF for the double-layer NF filter increased to 25.3%, 22.3%, and 20.5% for PM filtering with 0.3, 0.5, and 1.0 μm sizes, respectively, comparing with the discharged state of the filter membrane. The QF for the triple-layer NF filter decreases by 5.3%, 9.0%, and 6.4% comparing to the double-layer NF of it at PM 0.3, 0.5, and 1.0, respectively. The QF results show that the formation of electrostatic charge for fine PM filtration is reduced under periodic air blowing conditions at normal breathing levels. From the QF results, the QF improvement of the double-layer NF filter by cyclic air blowing is more effective than the triple-layer NF of it. The power of the airflow is not sufficient to cause bending of the filter structure owing to the increased number of mesh supporting layers, where the 1.2 times thicker membrane is required more bending stress to make the same deformation of a mask.

The trend of *η* after continuous dry or humid air penetrated a filter membrane is analyzed to evaluate the electrostatic charge sustainability. In continuous filter penetration, NF layers in the filter membrane are rearranged by airflow momentum, which affects the flow resistance of the NF membrane. The pressure drop decreased on the filter membranes after dry and humid air penetration (Appendix A). In humid airflow conditions, the pressure drop is reduced to 4.4% in air flow on the filter membrane comparing to the dry air flow condition. Decreasing of the filtration efficiency is not significant with the increase in the number of NF layers. In Figure 5c, the *η* result shows the persistence of electrostatic charges on the filter after continuous dry or humid air penetration. For the *η* at PM 2.5, the inertial impaction filtration on a multilayer NF filter membrane provides advantages to supporting the PVDF NF structure on the mesh layer.

As the size of the PM decreases, *η* also decreases because of the absence of electrostatic forces on the NF surface. In the filter structure of triple- or more NF layers, the structure shows the ability to preserve electrostatic charge compared to the filtration efficiency of a single-layer. In particular, the multilayer NF structure of the filter membrane is beneficial for retaining the electrostatic charges from the *η* at PM 0.3 and 0.5 filtration. From the filtration performance results, there is a trade-off between the retention and generation of electrostatic charges caused by the structure and deformability of the filter membrane, respectively. From the results, the triple-layer NF filter is suitable to apply a face mask filter maintaining high filtration performance, when considering the membrane deformability for tribo/piezoelectric charge generation in a humid air-breathing environment.

## 4. Conclusions

In this study, we demonstrate an electrostatic air filter membrane applied to a face mask with characteristics of electrostatic charge generation and retention in a humid air-breathing environment. The filter membrane has a structure in which layers of the nylon mesh and the PVDF NF are sequentially stacked. The layer-by-layer stacking PVDF NF and the nylon mesh layers has significant advantages for self-charging characteristics, which are caused by the triboelectricity in the polymer interface and the piezoelectricity in polar crystal of PVDF NF. As the number of NF layers increases, the filtration is most effective to PM 0.3 despite the same amount of the PVDF NF material. The filtration efficiency of the quintuple-layer NF filter is 99.5% for PM 0.3 filtration at an air velocity of 18.6 cm s^−1^, and a pressure drop of 200 Pa. As results of the face mask filter performance analysis, generation and retention of the electrostatic charges are sustained by filter bending through the human breathing cycle with a humid air penetration condition. In this point, compared to the single-layer NF filter membrane, a triple-layer NF filter shows optimal filtration performance with a QF of 0.019 Pa^−1^ for electrostatic charge generation. Even in electrical discharging by humid and dry air penetration environment, a triple-layer NF filter was better at maintaining filtration efficiency. The results indicate that the PVDF NF-nylon mesh multilayer structure would be a self-charging electrostatic filter, especially in fine PM filtration for face mask application.

## Figures and Tables

**Figure 1 polymers-13-03235-f001:**
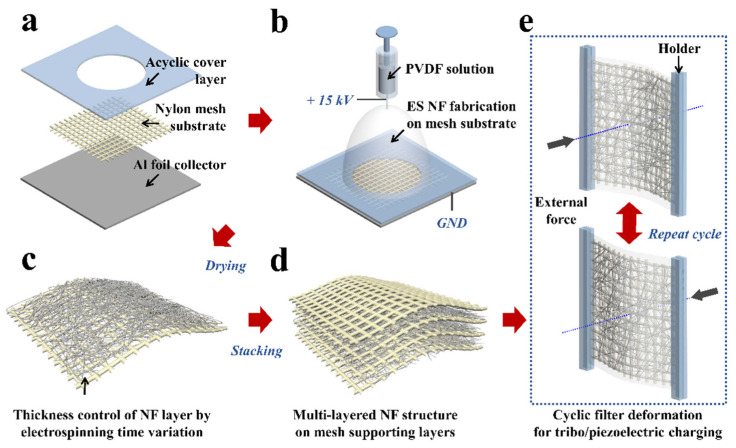
Schematics of the multilayer filter fabrication and electrostatic charging process. (**a**,**b**) PVDF NF membrane fabrication on a nylon mesh substrate by the electrospinning method. (**c**,**d**) Fabrication of a triple-layer filter by stacking layers of the PVDF NF membrane-nylon mesh. (**e**) Generation of electrostatic charges on the filter by the tribo/piezoelectric effect between nylon and PVDF materials in cyclic external force.

**Figure 2 polymers-13-03235-f002:**
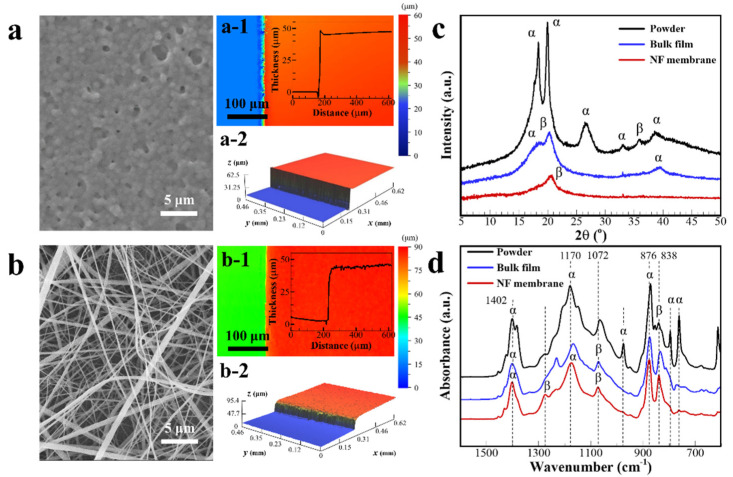
Surface geometry analysis of (**a**) bulk PVDF film and (**b**) electrospun PVDF NF membrane using SEM images and 3D optical analyzer. (**c**,**d**) XRD and FT-IR analysis results for PVDF with NF membrane, bulk film, and powder geometries.

**Figure 3 polymers-13-03235-f003:**
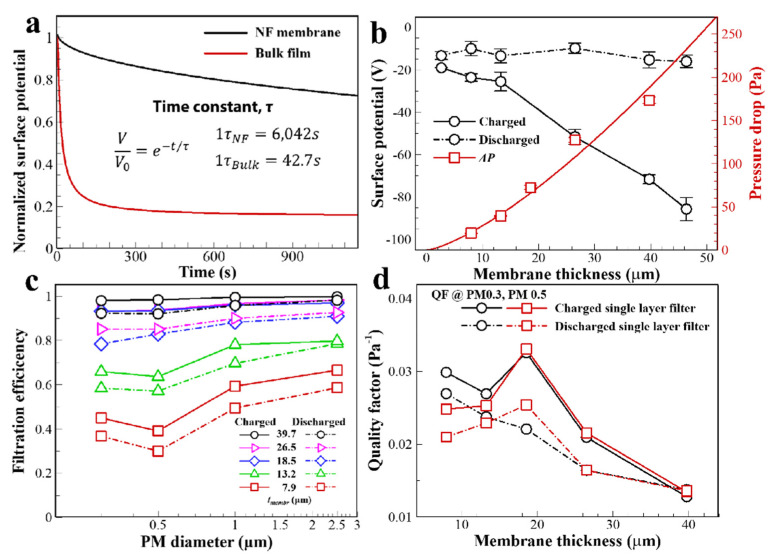
(**a**) Normalized surface potential decay of the PVDF bulk film and the PVDF NF membrane. (**b**) Pressure drop and surface potential of PVDF NF membrane of a single-layer filter by electrostatic charging and discharging of the PVDF NF filter as a function of NF membrane thickness. (**c**) Filtration efficiency of the filter at various PM sizes depending on the surface charging and discharging conditions. (**d**) QF of the single-layer filter as a function of NF membrane thickness and electrostatic charging state.

**Figure 4 polymers-13-03235-f004:**
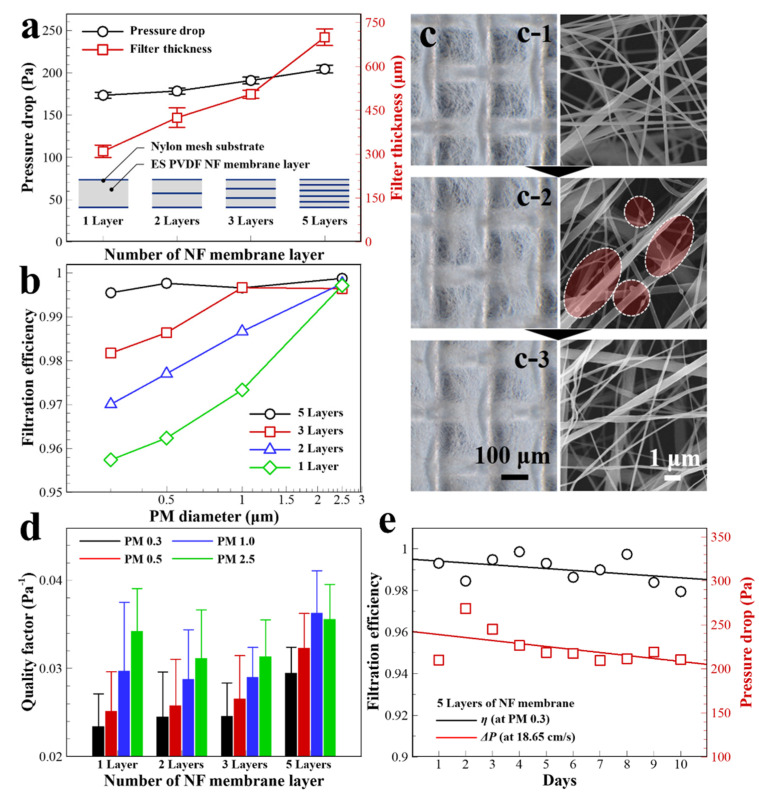
(**a**) Pressure drop of the multilayer NF filter depending on the layer composition. (**b**) Filtration efficiency of the number of layers in the multilayer NF filter for fine PM filtration. (**c**) OM and SEM images of the multilayer NF filter surface (**c-1**) before filtration, (**c-2**) after filtration, and (**c-3**) regeneration by oven drying after ethanol dipping. (**d**) Characteristics of averaged QF at different PM sizes by surface charging after regeneration each day for 10 days. (**e**) Filtration efficiency and pressure drop variation of a quintuple-layer of NF membrane filter for 10 days.

**Figure 5 polymers-13-03235-f005:**
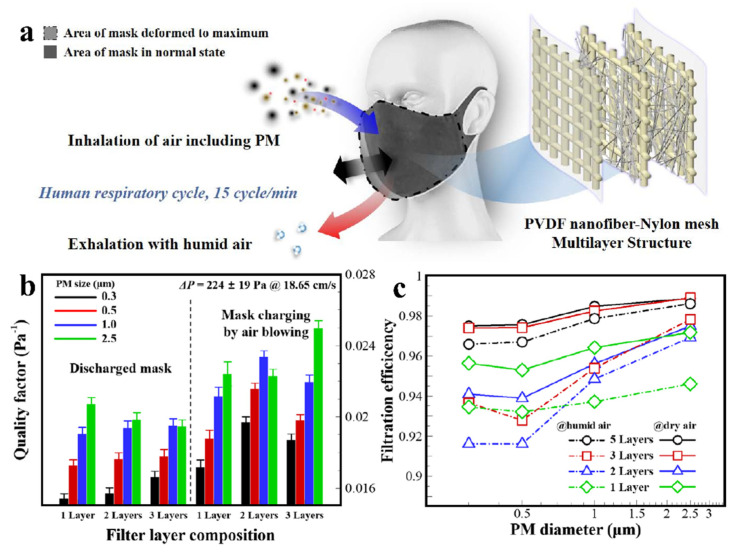
(**a**) Schematic for filter charging via bending and fluttering by air blowing, mimicking human respiration. (**b**) QF of the melt-blown PP layer-covered multilayer NF filter depending on discharged and charged states. (**c**) Persistence of filtration efficiency of the multilayer NF filter after continuous dry or humid air penetration for 0.5 h.

## Data Availability

The data presented in this study are available on request from the corresponding author.

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
