# Peer review of "Electrostatic Charge Retention in PVDF Nanofiber-Nylon Mesh Multilayer Structure for Effective Fine Particulate Matter Filtration for Face Masks"

_polymers, 2021, doi:10.3390/polym13193235_

Round 1

Reviewer 1 Report

Dear Authors,

Your manuscript on the " Electrostatic Charge Retention in PVDF Nanofiber-Nylon Mesh Multilayer Structure for Effective Fine Particle Matter Filtration for Face Masks" is highly relevant these days. This study is well documented, characterized, and possible for scale-up production. However, there is always room for improvement. 

I will not include in my comments those that involve sentence reconstructions and grammar. However, please improve your writing by not making too long sentences. Please let someone proofread your manuscript too.

 I hope you can address some of the comments and suggestions in your manuscript.

  1.  Please include more literature on the current studies of face masks using nanofiber membranes, even the ones not using electrospinning method. 
  2. The novelty of this study and its product is not well emphasized. I think you should include it in the last paragraph of your Introduction. You should convince the readers how this new type of face masks answered the new challenge for the fine particle filtration problem.
  3. How sustainable is the self-powdered electrostatically charged face mask once it is used? 
  4. What can you tell about the reusability of this kind of face mask?
  5. Is this type of facemask potential to replace the existing ones in the market? Or is the application of this face mask only selective to a particular type of people or circumstances, considering the materials' cost.

Thank you

Author Response

Manuscript ID: polymers-1359955

"Electrostatic Charge Retention in PVDF Nanofiber-Nylon Mesh Multilayer Structure for Effective Fine Particulate Matter Filtration for Face Masks" by Dong Hee Kang, Na Kyong Kim, Hyun Wook Kang *

We appreciate the referee for her/his helpful and stimulating comments. We have prepared a revised manuscript in accord with the comments of the referee. The revised paragraph, and figures have been inserted into the text, and some typos have been corrected. Please see the revised manuscript. As to the specific responses in the revised paper, we would like to note the following modifications.

Dear Authors,

Your manuscript on the "Electrostatic Charge Retention in PVDF Nanofiber-Nylon Mesh Multilayer Structure for Effective Fine Particle Matter Filtration for Face Masks" is highly relevant these days. This study is well documented, characterized, and possible for scale-up production. However, there is always room for improvement.

I will not include in my comments those that involve sentence reconstructions and grammar. However, please improve your writing by not making too long sentences. Please let someone proofread your manuscript too.

I hope you can address some of the comments and suggestions in your manuscript.

(1) Please include more literature on the current studies of face masks using nanofiber membranes, even the ones not using electrospinning method.

  • As mentioned by referee, we added the related references with face mask in the "Introduction section" section in line 33. The added references are present as below,
  • [4]Tebyetekerwa, M.; Xu, Z.; Yang, S.; and Ramakrishna, S. Electrospun nanofibers-based face masks. Adv. Fiber Mater., 2020, 2, 161-166., [5] Essa, W. K.; Yasin, S. A.; Saeed, I. A.; and Ali, G. A. Nanofiber-Based Face Masks and Respirators as COVID-19 Protection: A Review. Membranes, 2021, 11, 250., [6] Zhang, Z.; Ji, D.; He, H.; and Ramakrishna, S. Electrospun ultrafine fibers for advanced face masks. Mater. Sci. Eng. R Rep., 2021, 143, 100594., and [7] Ullah, S.; Ullah, A.; Lee, J.; Jeong, Y.; Hashmi, M.; Zhu, C.; Joo, K.I.; Cha, H.J.; and Kim, I. S. Reusability comparison of melt-blown vs nanofiber face mask filters for use in the coronavirus pandemic. ACS Appl. Nano Mater., 2020, 3, 7231-7241.
  • Also, the last paragraph of the "Introduction" section has been modified with current discussing the electrospinning technique and the applications of nanofibers, " A filter membrane is fabricated by electrospinning of the PVDF on a nylon mesh layer, used as a NF support structure. The fine PM filtration efficiency of the filter is compared depending on the electrostatic charge generation and retention characteristics via tribo/piezoelectricity. The filter membrane stacks are composed of the nylon mesh layers with the one to two, three, and five layers of electrospun PVDF NF, respectively. It has a structure in which the nylon mesh and PVDF NF layers are stacked in order. The number of nylon mesh layers of filter membrane is adjusted for enhancing the PM filtration rate and diminishing the low pressure drop. As increasing the number of the mesh layers in the filter membrane, it is effective to filtration performance as the PM size decreasing. A triple-layer NF filter structure has enhanced electrostatic charge generation on the filter in a respiration mimicking environment, which has advantages for a face mask application. A new type of face mask is possible to generate electrostatic charges in the breathing process after wearing a face mask via the unique multilayer structure of the filter, which overcame the limitation to fine PM filtration performance on the face mask.", in lines 74-88.

(2) The novelty of this study and its product is not well emphasized. I think you should include it in the last paragraph of your Introduction. You should convince the readers how this new type of face masks answered the new challenge for the fine particle filtration problem.

  • As mentioned by referee, a sentence is added to emphasize the novelty of this study at the end of the last paragraph in the "Introduction" section as follows, "A new type of face mask is possible to generate electrostatic charges in the breathing process after wearing a face mask via the unique multilayer structure of the filter, which overcame the limitation to fine PM filtration performance on the face mask.", in lines 85-88.

(3) How sustainable is the self-powdered electrostatically charged face mask once it is used?

  • The PM filtration efficiency present depending on the electrostatic force retention characteristics between the discharged state (only mechanical filtration efficiency) and the fully charged state (mechanical and electrostatic filtration efficiency).
  • If there is no physical deformation of the filter, the PM filtration efficiency will maintain between 96.5 and 99.5% of the PM 0.3 at the 5-layer filter, as shown in Figure 4b and Figure 5c. Therefore, based on the normalized surface potential in equation 3a, the filtration efficiency of PM 0.3 takes 69.6 min to decrease to 98% in a 5-layer filter membrane.

(4) What can you tell about the reusability of this kind of face mask?

  • Compared with the mechanical filtration of the mask filter, the fine PM-adhered nanofiber surface can be easily cleaned through the loss of electrostatic force. In which electrostatic force is regenerated by surface friction, the filter presents high PM filtration efficiency again, and this reusability of filter can minimize waste of resources.

(5) Is this type of facemask potential to replace the existing ones in the market? Or is the application of this face mask only selective to a particular type of people or circumstances, considering the materials' cost.

Yes. It increases the user's risk of exposure to fine particles. The filtration performance of the fine PM is decreases dramatically when the disposable face mask is used multiple times. In this point of view, depending on the result of the Figure 4e, it can be used continuously more than 10 times by the electrostatic charge generation on the tribo/piezoelectricity of the filter. Also, it is possible to reduce the waste of resources used in the filter.

Reviewer 2 Report

In this study, a novel face mask with an electrostatically rechargeable filter membrane is fabricated and tested. This electrostatically rechargeable filter membrane is formed by stacking the electrospun PVDF nanofibre mat on a nylon fabric layer by layer. The paper is interesting for the scientific community and up to date and can be recommended for publication after some revision. 

The introduction part should be expanded a little with current sources discussing the electrospinning technique and the applications of nanofibres.

Lines 47-49: Two statements are made and groups are mentioned, but the sources are missing. Please insert sources at this point. 
Line 82: Please specify exactly the nylon mesh substrate used, such as weight g/m2 and how many threads per cm.
Line 85: How exactly was the nylon mesh washed with acetone, ethanol and deionised water? Under what parameters?
Line 87: The exact manufacturer, model etc. are missing for the convection oven, electrospinning machine as well as the detailed electrospinning parameters. Some parameters are given from line 135 onwards, but not completely. Also it is not entirely clear how many samples were produced and how they are constructed.

A sample overview should be presented in the Materials and Methods part for a clear overview. It is clear later in the manuscript which samples were prepared and how they are constructed, but at the beginning it is a bit confusing because the exact overview is missing. 
Line 132: It is very difficult to see graphs a-2 and b-2 because they are too small and hardly readable. 
Line 289: At this point it is said that NF membrane surface of the multilayer filter before and after PM filtration is immersed in ethanol and dried, under which parameters etc. was it done?

Author Response

Manuscript ID: polymers-1359955

"Electrostatic Charge Retention in PVDF Nanofiber-Nylon Mesh Multilayer Structure for Effective Fine Particulate Matter Filtration for Face Masks" by Dong Hee Kang, Na Kyong Kim, Hyun Wook Kang *

We appreciate the referee for her/his helpful and stimulating comments. We have prepared a revised manuscript in accord with the comments of the referee. The revised paragraph, and figures have been inserted into the text, and some typos have been corrected. Please see the revised manuscript. As to the specific responses in the revised paper, we would like to note the following modifications.

In this study, a novel face mask with an electrostatically rechargeable filter membrane is fabricated and tested. This electrostatically rechargeable filter membrane is formed by stacking the electrospun PVDF nanofibre mat on a nylon fabric layer by layer. The paper is interesting for the scientific community and up to date and can be recommended for publication after some revision.

(1) The introduction part should be expanded a little with current sources discussing the electrospinning technique and the applications of nanofibres.

  • The last paragraph of the "Introduction" section has been modified with current discussing the electrospinning technique and the applications of nanofibers, " A filter membrane is fabricated by electrospinning of the PVDF on a nylon mesh layer, used as a NF support structure. The fine PM filtration efficiency of the filter is compared depending on the electrostatic charge generation and retention characteristics via tribo/piezoelectricity. The filter membrane stacks are composed of the nylon mesh layers with the one to two, three, and five layers of electrospun PVDF NF, respectively. It has a structure in which the nylon mesh and PVDF NF layers are stacked in order. The number of nylon mesh layers of filter membrane is adjusted for enhancing the PM filtration rate and diminishing the low pressure drop. As increasing the number of the mesh layers in the filter membrane, it is effective to filtration performance as the PM size decreasing. A triple-layer NF filter structure has enhanced electrostatic charge generation on the filter in a respiration mimicking environment, which has advantages for a face mask application. A new type of face mask is possible to generate electrostatic charges in the breathing process after wearing a face mask via the unique multilayer structure of the filter, which overcame the limitation to fine PM filtration performance on the face mask.", in lines 74-88.
  • Also, we added the related references with face mask in the "Introduction section" section in line 33. The added references are present as below,
  • [4]Tebyetekerwa, M.; Xu, Z.; Yang, S.; and Ramakrishna, S. Electrospun nanofibers-based face masks. Adv. Fiber Mater., 2020, 2, 161-166.
  • [5] Essa, W. K.; Yasin, S. A.; Saeed, I. A.; and Ali, G. A. Nanofiber-Based Face Masks and Respirators as COVID-19 Protection: A Review. Membranes, 2021, 11, 250.
  • [6] Zhang, Z.; Ji, D.; He, H.; and Ramakrishna, S. Electrospun ultrafine fibers for advanced face masks. Mater. Sci. Eng. R Rep., 2021, 143, 100594.
  • [7] Ullah, S.; Ullah, A.; Lee, J.; Jeong, Y.; Hashmi, M.; Zhu, C.; Joo, K.I.; Cha, H.J.; and Kim, I. S. Reusability comparison of melt-blown vs nanofiber face mask filters for use in the coronavirus pandemic. ACS Appl. Nano Mater., 2020, 3, 7231-7241.
  •  

(2) Lines 47-49: Two statements are made and groups are mentioned, but the sources are missing. Please insert sources at this point.

  • As mentioned by referee, the references [14] and [15] are inserted at the end of the sentences as below, "One research group suggested an aluminum coated polyester microfiber fabrication method for a high-performance electrostatic filtration system [14]. Another research group presented a copper layer-coated polyacrylonitrile microfiber for PM filtration with a de-creasing carbon monoxide concentration [15].", in lines 46-49.

(3) Line 82: Please specify exactly the nylon mesh substrate used, such as weight g/m2 and how many threads per cm.

  • In the manuscript, the nylon mesh substrate has 40 g m-2 ­­area density and 433 mesh threads per cm. The sentence is modified as below, "The nylon mesh substrate (woven Nylon 66A screen, Flon ind.) has a 161 μm pore size, 40 g m-2 area density and 433 cm-1 mesh threads." in the line 93-95.

(4) Line 85: How exactly was the nylon mesh washed with acetone, ethanol and deionised water? Under what parameters?

  • The nylon mesh is washed using an ultrasonic cleaner each 5 minutes in the acetone, ethanol and deionized water, sequentially. The sentence is modified as below, "The nylon mesh layer was cut into a 100 × 100 mm2 area and washed sequentially for 5 min each using acetone, ethanol, and deionized (DI) water in an ultrasonic cleaner.", in the line 97-99.

(5) Line 87: The exact manufacturer, model etc. are missing for the convection oven, electrospinning machine as well as the detailed electrospinning parameters. Some parameters are given from line 135 onwards, but not completely. Also it is not entirely clear how many samples were produced and how they are constructed.

  • The manufactural and model of the convection oven are added in the lines 99-100, and the sentence is modified as follow, " The washed mesh was dried in a convection oven (OF-12PW, Jeio tech) for 12 h at 60°C.".
  • The nanofiber fabrication is performed by handmade electrospinning apparatus. Therefore, we added manufactural and model of a high voltage source as well as the detailed electrospinning parameters. The revised sentence is presented as follows, " A high voltage source (B.150, KSC Korea switching) is used to occur the electrical potential gradient (15 kV) on the aluminum foil collector leading from the metal needle tip (needle tip-to-collector distance, 150 mm). The Figure 1b shows that the positively charged PVDF solution (flow rate, 30 μL min-1) stretches to the NF geometry and randomly piled up on the mesh. Depending on the fabrication conditions, each membrane was prepared by 10 samples.", in lines 154-162.

(6) A sample overview should be presented in the Materials and Methods part for a clear overview. It is clear later in the manuscript which samples were prepared and how they are constructed, but at the beginning it is a bit confusing because the exact overview is missing.

  • At the beginning of the "Materials and Methods" section, we increased contents of a first paragraph to help readers understand our mask filter structure as follow, " For the membrane of filter fabrication, electrospinning method is performed using PVDF solution on the nylon mesh substrate. The multilayer structure of the face mask filter is composed of the stacked electrospun PVDF NF-nylon mesh substrates.", in lines 91-93.

(7) Line 132: It is very difficult to see graphs a-2 and b-2 because they are too small and hardly readable.

  • As mentioned by referee, the Figure 2a-1,2 and b-1,2 are modified to see clearly in line 148.

(8) Line 289: At this point it is said that NF membrane surface of the multilayer filter before and after PM filtration is immersed in ethanol and dried, under which parameters etc. was it done?

In Figure 4c to show the filter washing and drying process, the detailed explanations are added in the sentences as below, "In Figure 4c, optical microscopy (OM) and SEM images show the NF membrane surface of the multilayer filter (Figure 4c-1) before and (Figure 4c-2) after PM filtration, and after ethanol dipping (completely immersing for 5 min) and drying (in a convection oven for 12 h at 40°C) of the filter.", in lines 314-317.

Reviewer 3 Report

The reviewed manuscript investigates the enhancement ability of the electrostatic charge generation for their new fabricated face mask filters by using PVDF nanofiber layers with nylon mesh. The article is made at a good scientific and technical level, and topic is interesting. In order to improve the readability and clarity of the manuscript, some major concerns need to be addressed before the paper is to be processed ahead:

  1. The abstract is a mini version of manuscript that proceeds. So, include introduction, methodology, results and concluding remarks in a precise but effective manner.
  2. Why all “figure” and “table” words are followed by the letter “S” when cited in the text?
  3. More details about electrospinning process is needed (needle diameter, space between needle and collector, solution flowrate, …..).
  4. L97: “Filtration performance measurement was performed in a chamber as shown pneumatic symbol in Figure S2. An incense burning was performed for a PM source in the first chamber.” Sentence is not clear and no relation with figure 2. Please reconstruct the phrase carefully.
  5. L98 – 104: schematic layout is needed for the described system.
  6. L105 – 106: The experimental apparatus for the humid air is not presented in figure 3 as mentioned, and no relation with the provided figure 3.
  7. L109: “The details of the apparatus are described in Figure S4.” Again, the cyclic air blowing apparatus is not related to figure 4 as mentioned.
  8. L110: “…. composed with the PP layer-covered filter membrane … ” please define any abbreviations firstly before using in the text.
  9. L108: Table S1 is missing in the manuscript.
  10. L112: Table S2 is missing in the manuscript.
  11. L97: “…. patterns of the PVDF with NF membrane, bulk and powder.” What authors mean by powder?
  12. L134, Figure 2 caption: “(c, d) FT-IR and XRD analysis results for PVDF …..” It is opposite, (c) is XRD and (d) is FTIR.
  13. Figure 2: What is the benefit of performing SEM for the PVDF film if authors used it in their study in the nanofiber form? I recommend to remove figure 2.a totally, plus the bulk film pattern from (c) and (d) .
  14. Figure 1.a: why adding the cyclic cover layer on the Nylon mesh during electrospinning?
  15. Figure 2.c (XRD): What is alpha (a) and beta (β)? Please identify all peaks in each pattern correctly on the same graph and provide the unique powder diffraction file (PDF) for each element/compound in the pattern.
  16. L162: Table S3 is missing in the manuscript.
  17. L280: Table S4 is missing in the manuscript.
  18. L379: Supplementary Materials are not found on the specified link “Error 404 - File not found”.
  19. Discussion is lake of scientific explanation for the obtained results. Authors should attribute the results achieved to a clear scientific reason.
  20. Figure 4.a: legend is missing.

Author Response

"Electrostatic Charge Retention in PVDF Nanofiber-Nylon Mesh Multilayer Structure for Effective Fine Particulate Matter Filtration for Face Masks" by Dong Hee Kang, Na Kyong Kim, Hyun Wook Kang *

We appreciate the referee for her/his helpful and stimulating comments. We have prepared a revised manuscript in accord with the comments of the referee. The revised paragraph, and figures have been inserted into the text, and some typos have been corrected. Please see the revised manuscript. As to the specific responses in the revised paper, we would like to note the following modifications.

The reviewed manuscript investigates the enhancement ability of the electrostatic charge generation for their new fabricated face mask filters by using PVDF nanofiber layers with nylon mesh. The article is made at a good scientific and technical level, and topic is interesting. In order to improve the readability and clarity of the manuscript, some major concerns need to be addressed before the paper is to be processed ahead:

(1) The abstract is a mini version of manuscript that proceeds. So, include introduction, methodology, results and concluding remarks in a precise but effective manner.

  • As mentioned by referee, the beginning of the introduction part, we added references the related researches with the mask filter in the "Introduction section" section in line 33. The added references are present as below, [4]Tebyetekerwa, M.; Xu, Z.; Yang, S.; and Ramakrishna, S. Electrospun nanofibers-based face masks. Adv. Fiber Mater., 2020, 2, 161-166., [5] Essa, W. K.; Yasin, S. A.; Saeed, I. A.; and Ali, G. A. Nanofiber-Based Face Masks and Respirators as COVID-19 Protection: A Review. Membranes, 2021, 11, 250., [6] Zhang, Z.; Ji, D.; He, H.; and Ramakrishna, S. Electrospun ultrafine fibers for advanced face masks. Mater. Sci. Eng. R Rep., 2021, 143, 100594., and [7] Ullah, S.; Ullah, A.; Lee, J.; Jeong, Y.; Hashmi, M.; Zhu, C.; Joo, K.I.; Cha, H.J.; and Kim, I. S. Reusability comparison of melt-blown vs nanofiber face mask filters for use in the coronavirus pandemic. ACS Appl. Nano Mater., 2020, 3, 7231-7241.
  • Also, the last paragraph of the "Introduction" section has been modified with current discussing the electrospinning technique and the applications of nanofibers, " A filter membrane is fabricated by electrospinning of the PVDF on a nylon mesh layer, used as a NF support structure. The fine PM filtration efficiency of the filter is compared depending on the electrostatic charge generation and retention characteristics via tribo/piezoelectricity. The filter membrane stacks are composed of the nylon mesh layers with the one to two, three, and five layers of electrospun PVDF NF, respectively. It has a structure in which the nylon mesh and PVDF NF layers are stacked in order. The number of nylon mesh layers of filter membrane is adjusted for enhancing the PM filtration rate and diminishing the low pressure drop. As increasing the number of the mesh layers in the filter membrane, it is effective to filtration performance as the PM size decreasing. A triple-layer NF filter structure has enhanced electrostatic charge generation on the filter in a respiration mimicking environment, which has advantages for a face mask application. A new type of face mask is possible to generate electrostatic charges in the breathing process after wearing a face mask via the unique multilayer structure of the filter, which overcame the limitation to fine PM filtration performance on the face mask.", in lines 74-88.

(2) Why all “figure” and “table” words are followed by the letter “S” when cited in the text?

  • In the manuscript, the figures and tables following with the letter "S" are a head letter of the "Supplementary Material". All of the sentences including the supplementary figures and tables are revised in order to describe the meaning clearly as below,
  • Lines 95-97: The nylon mesh supporting layer is almost no affect to increase the airflow (~0.5 Pa per nylon mesh layer at 18.64 cm s-1 air velocity) resistance (Figure S1, Supplementary Material).
  • Lines 111-113: Filtration performance measurement was performed in a chamber. (The pneumatic symbol in Figure S2, Supplementary Material).
  • Lines 120-121: The experimental apparatus was prepared for the electrical discharging of the filter membrane via humid air penetration (Figure S3, Supplementary Material).
  • Lines 121-123: The moisture concentration in the supplied air was calculated by the average absolute humidity per minute as a level of typical breathing for an adult (Table S1, Supplementary Material).
  • Lines 123-125: In the mask filtration test, the filter membrane is electrically charged by cyclic air blowing (Figure S4, Supplementary Material).
  • Lines 127-128: The power of airflow was calculated based on the maximum flow rate and pressure difference of typical breathing for a healthy adult (Table S2, Supplementary Material).
  • Lines 183-186: The weight measurements of the bulk film per unit area are compared by multiplying the PVDF density and the average thickness value from the surface profiler (Table S3, Supplementary Material); a difference of less than 1.8% is observed.
  • Lines 187-190: In Figure 2b-1 and Figure 2b-2, the membrane layer shows entangled NFs of various diameters (260.50±148.80 nm, Figure S5, Supplementary Material) forming a porous structure with 44.48±1.02 μm thickness.
  • Lines 232-234: The thickness of the PVDF NF membrane was controlled by varying the electrospinning time, and a linear correlation was observed between them (Figure S6, Supplementary Material).
  • Lines 269-270: Figure S7 (in Supplementary Material) shows the QFs for various NF membrane thicknesses and charge states for the single-layer filter depending on the PM size.
  • Lines 304-306: Compared with the performance of commercial face mask filters (Table S4, Supplementary Material), the multilayer filter showed a considerably higher performance in terms of the ΔP and η characteristics.
  • Lines 362-363: The pressure drop decreased on the filter membranes after dry and humid air penetration (Figure S8, Supplementary Material).

(3) More details about electrospinning process is needed (needle diameter, space between needle and collector, solution flowrate, …..).

  • The detailed electrospinning parameters are added in the sentence as follows, "A high voltage source (B.150, KSC Korea switching) is used to occur the electrical potential gradient (15 kV) on the aluminum foil collector leading from the metal needle tip (needle tip-to-collector distance, 150 mm). The Figure 1b shows that the positively charged PVDF solution (flow rate, 30 μL min-1) stretches to the NF geometry and randomly piled up on the mesh. Depending on the fabrication conditions, each membrane was prepared by 10 samples.", in lines 154-162.

(4) L97: “Filtration performance measurement was performed in a chamber as shown pneumatic symbol in Figure S2. An incense burning was performed for a PM source in the first chamber.” Sentence is not clear and no relation with figure 2. Please reconstruct the phrase carefully.

  • The Figure S2 is described in the "Supplementary Material" to show experimental setup briefly. And the sentence is revised in order to describe the meaning clearly as below, "Incense was used for the PM source in the first chamber. ", in line 113.

(5) L98 – 104: schematic layout is needed for the described system.

  • The brief experimental setup is described in the Supplementary Material. The sentence including the supplementary figure is revised in order to describe the meaning clearly as below, Lines 111-113: Filtration performance measurement was performed in a chamber. (The pneumatic symbol in Figure S2, Supplementary Material).

(6) L105 – 106: The experimental apparatus for the humid air is not presented in figure 3 as mentioned, and no relation with the provided figure 3.

  • The brief experimental setup using pneumatic symbol to show humid air test is described in Figure S3 of the Supplementary Material. The sentence including the supplementary figure is revised in order to describe the meaning clearly as below, Lines 120-121: The experimental apparatus was prepared for the electrical discharging of the filter membrane via humid air penetration (Figure S3, Supplementary Material).

(7) L109: “The details of the apparatus are described in Figure S4.” Again, the cyclic air blowing apparatus is not related to figure 4 as mentioned.

  • The detailed experimental apparatus for the electrical charging of the filter membrane is described in Figure S4 of the Supplementary Material. The sentence including the supplementary figure is revised in order to describe the meaning clearly as below, Lines 123-125: In the mask filtration test, the filter membrane is electrically charged by cyclic air blowing (Figure S4, Supplementary Material).

(8) L110: “…. composed with the PP layer-covered filter membrane … ” please define any abbreviations firstly before using in the text.

  • We described the full name of the polypropylene which abbreviation is a "PP" in line 110, and the sentence is modified as follows, "The mask is composed with the polypropylene (PP) layer-covered filter membrane, which was deformed by cyclic air blowing at 15 cycles min-1."

(9) L108: Table S1 is missing in the manuscript.

  • The sentences including the supplementary table data are revised in order to describe the meaning clearly as below, Lines 121-123: The moisture concentration in the supplied air was calculated by the average absolute humidity per minute as a level of typical breathing for an adult (Table S1, Supplementary Material).

(10) L112: Table S2 is missing in the manuscript.

  • The sentences including the supplementary table data are revised in order to describe the meaning clearly as below, Lines 126-127: The power of airflow was calculated based on the maximum flow rate and pressure difference of typical breathing for a healthy adult (Table S2, Supplementary Material).

(11) L97: “…. patterns of the PVDF with NF membrane, bulk and powder.” What authors mean by powder?

  • The powder is a neat PVDF powder to compare with the PVDF nanofiber membrane by the spectroscopic analysis. In Figure 2c and d, the XRD and FT-IR analysis results are including the diffraction pattern and spectroscopic diagram of the neat powder, which served as a good control for the distinction between α and β phases in our membrane.

(12) L134, Figure 2 caption: “(c, d) FT-IR and XRD analysis results for PVDF …..” It is opposite, (c) is XRD and (d) is FTIR.

  • As mentioned by referee, the order of the figure caption 2c and 2d is changed correctly. The Figure 2 caption in line 147 is modified as follows, "(c, d) XRD and FT-IR analysis results for PVDF with NF membrane, bulk film, and powder geometries.".

(13) Figure 2: What is the benefit of performing SEM for the PVDF film if authors used it in their study in the nanofiber form? I recommend to remove figure 2.a totally, plus the bulk film pattern from (c) and (d).

  • In Figure 2, we intended to show physical characteristics of the PVDF nanofiber, and the geometrical and analytical results about the film having the same volume (the bulk film) had been added for comparison.
  • In Table S3, the geometrical analysis result presents the structural characteristics (porosity), and the superiority of the porous structure related with the electrostatic retention characteristics shown in Figure. 3a.

(14) Figure 1.a: why adding the cyclic cover layer on the Nylon mesh during electrospinning?

  • During the electrospinning process, acrylic cover layer is working to insulation layer in the electric fields. As shown in Figure 1b, the electrospun nanofiber is formed in a circular shape on the mesh substrate, which is not covered with the acrylic layer. The above sentence has been added in the manuscript in lines 157-159.

(15) Figure 2.c (XRD): What is alpha (a) and beta (β)? Please identify all peaks in each pattern correctly on the same graph and provide the unique powder diffraction file (PDF) for each element/compound in the pattern.

  • In the manuscript, the α and β are crystalline domains, which they are formed by a mixture of TGTG' (Trans-Gauche-Trans-Gauche conformations) chains and all-TTTT trans planar zigzag chains, respectively. The above sentence has been added in the manuscript in lines 192-194.
  • We had been identified the peak position in the figure 2c and d (XRD and FT-IR analysis graph) and the describe the detailed the locations of Bragg-peak scattering angles (from XRD) and the molecules absorb specific frequencies (from FT-IR) in the 2nd paragraph of the "3.2 Geometric and spectroscopic analysis" section.
  • Also, the XRD and FT-IR analysis results are including the diffraction pattern and spectroscopic diagram of the neat powder, which served as a good control for the distinction between α and β phases in our membrane.

(16) L162: Table S3 is missing in the manuscript.

  • The sentences including the supplementary table data are revised in order to describe the meaning clearly as below, Lines 183-186: The weight measurements of the bulk film per unit area are compared by multiplying the PVDF density and the average thickness value from the surface profiler (Table S3, Supplementary Material); a difference of less than 1.8% is observed.

(17) L280: Table S4 is missing in the manuscript.

  • The sentences including the supplementary table data are revised in order to describe the meaning clearly as below, Lines 304-306: Compared with the performance of commercial face mask filters (Table S4, Supplementary Material), the multilayer filter showed a considerably higher performance in terms of the ΔP and η characteristics.

(18) L379: Supplementary Materials are not found on the specified link “Error 404 - File not found”.

  • As mentioned by referee, we did re-upload the "supplementary material" again.

(19) Discussion is lake of scientific explanation for the obtained results. Authors should attribute the results achieved to a clear scientific reason.

  • As mentioned by referee, the "Results" section is changed to "Results and Discussion", which had been included the discussion of the experimental results at the end of the sub-section from "3.1. Materials for Electrostatic Filter" to "3.4. Evaluation of Filtration Performance".

(20) Figure 4.a: legend is missing.

  • In figure 4a, we added a legend in order to describe clearly.

Round 2

Reviewer 2 Report

Dear authors, the topic you are presenting is very relevant and interesting. The manuscript has been well revised and can be recommended for publication. With best regards

Reviewer 3 Report

The revision is satisfactory and the authors have provided amendments to all the suggested queries. Therefore, I recommend this work for publication in Polymers Journal.